# A low-cost fluorescence reader for *in vitro* transcription and nucleic acid detection with Cas13a

**Florian Katzmeier**[1☯], **Lukas Aufinger**[1☯], **Aurore Dupin**[1☯], **Jorge Quintero**[2☯], **Matthias Lenz**[1], **Ludwig Bauer**[1], **Sven Klumpe**[1], **Dawafuti Sherpa**[2], **Benedikt Dürr**[2], **Maximilian Honemann**[1], **Igor Styazhkin**[1], **Friedrich C. Simmel**[1], **Michael Heymann**[3]*

**1** Physics Department and ZNN, Technical University of Munich, Garching, Germany, **2** Department of Biology, Ludwig-Maximilians-Universität Munich, Martinsried, Germany, **3** Intelligent Biointegrative Systems Group, Institute for Biomaterials and Biomolecular Systems, University Stuttgart, Germany

☯ These authors contributed equally to this work.
* michael.heymann@bio.uni-stuttgart.de

**Data Availability Statement:** All relevant data are within the paper and its Supporting Information files.

## Abstract

Point-of-care testing (POCT) in low-resource settings requires tools that can operate independently of typical laboratory infrastructure. Due to its favorable signal-to-background ratio, a wide variety of biomedical tests utilize fluorescence as a readout. However, fluorescence techniques often require expensive or complex instrumentation and can be difficult to adapt for POCT. To address this issue, we developed a pocket-sized fluorescence detector costing less than \$15 that is easy to manufacture and can operate in low-resource settings. It is built from standard electronic components, including an LED and a light dependent resistor, filter foils and 3D printed parts, and reliably reaches a lower limit of detection (LOD) of $\approx 6.8$ nM fluorescein, which is sufficient to follow typical biochemical reactions used in POCT applications. All assays are conducted on filter paper, which allows for a flat detector architecture to improve signal collection. We validate the device by quantifying *in vitro* RNA transcription and also demonstrate sequence-specific detection of target RNAs with an LOD of 3.7 nM using a Cas13a-based fluorescence assay. Cas13a is an RNA-guided, RNA-targeting CRISPR effector with promiscuous RNase activity upon recognition of its RNA target. Cas13a sensing is highly specific and adaptable and in combination with our detector represents a promising approach for nucleic acid POCT. Furthermore, our open-source device may be used in educational settings, through providing low cost instrumentation for quantitative assays or as a platform to integrate hardware, software and biochemistry concepts in the future.

## Introduction

Fluorescence is widely used to diagnose infectious diseases fast and with high sensitivity, for instance through nucleic acid detection via quantitative PCR (qPCR) or affinity based techniques such as ELISA [1]. Over the past decades, point-of-care testing (POCT) has matured to

**Funding:** Financial support was provided by TU Munich (through the TUM Board of Management and the Departments of Physics and Chemistry), Lehre@LMU, the Nanosystems Initiative München (NIM), the European Research Council (grant agreement no. 694410—AEDNA) and UnternehmerTUM. M. Heymann gratefully acknowledges support through the Joachim Herz Foundation. Reagents, consumables and services were in part provided as in-kind contributions by IDT, Biomers, New England Biolabs, GATC Biotech, Promega, Scienova, Eurofins Genomics, GE Healthcare Carl Roth, Quiagen.

**Competing interests:** The authors have declared that no competing interests exist.

efficiently manage medical conditions such as diabetes or pregnancy, where rapid on-site diagnosis or monitoring is preferred. Although many fluorescence-based diagnostic techniques are, in principle, suitable for the rapid detection and monitoring of disease outbreaks, their application in disaster and low resource scenarios has been hampered by the need for bulky and expensive laboratory infrastructure [2]. One approach to reduce assay size and costs is to utilize colorimetric readouts [3, 4]. Compared to such visual readouts, however, the higher signal-to-background ratio associated with fluorescence enables quantitative measurements with generally 10 to 100-fold higher sensitivity [5].

Accordingly, portable low-cost open-source fluorescence readers are an attractive option for transferring fluorescence based diagnostics to in-field applications [6]. Here, the challenge is to achieve a sufficiently high sensitivity, small size and low power consumption, while abandoning expensive high-grade optical components. This also means that the reduced cost is usually compromised with a higher degree of specialization, a lower degree of automation, and a lower sensitivity and throughput. Despite these trade-offs, many functioning low-cost fluorescence readers, summarized in Table 1, have been designed and built for various POCT applications, reaching sensitivities in the nanomolar range at material costs as low as ≈ $50. While nowadays LEDs are commonly used as inexpensive, low-power excitation light sources, the main limitations in terms of cost vs. sensitivity are the need for high quality filter sets, magnifying optics and optical sensors [12].

Another major challenge in POCT is sample preparation with limited laboratory infrastructure. In this context, paper has been successfully applied as a low-cost and versatile carrier material. It can be easily patterned with application specific microfluidic channels using a standard wax-based desktop printer [13]. Paper is compatible with many biochemical reaction assays, which can be freeze-dried onto paper test strips for room temperature storage and distribution [14]. Despite these benefits, fluorescence detection on paper is challenging, as its autofluorescence and strong light scattering can compromise sensitivity significantly [15].

Groundbreaking work towards in-field application of paper-based fluorescence detection was recently conducted by Pardee *et al.*, who developed a nucleic acid paper assay for the detection of infectious diseases such as the Ebola virus [16]. For this, a riboregulator-based RNA sensor (a 'toehold switch' [17]) was coupled to a colorimetric readout cascade in a cell-free protein synthesis (CFPS) system that could be distributed when lyophilized on filter paper. The sequence specificity of the system was further enhanced by incorporating CRISPR/Cas9, and isothermal amplification of the extracted RNA was applied to improve sensitivity [18]. Later, Gootenberg et al. developed a related detection assay based on recombinase polymerase amplification (RPA, a technique for isothermal amplification [19]) and CRISPR/Cas13a, termed SHERLOCK [20]. SHERLOCK exploits the RNA-guided RNase activity of Cas13a, which upon target binding develops a collateral RNase activity [21, 22] that could be monitored with a fluorescent readout. More recently, a similar feature was also found in

**Table 1. Comparison of low-cost fluorescence detectors.** All parameters are noted as specified by the authors, or were estimated, as indicated by '≈' or 'n.s.' (not specified) if not enough information was available. Other abbreviations, 'x': available detection channels, 'HV': high voltage supply needed for capillary electrophoresis.

| Assay demonstrated | Sensitivity (LOD) | Cost | Size | Power/Voltage | x | Ref. |
|---|---|---|---|---|---|---|
| Capillary electrophoresis | 920 nM Fluorescein | $47 | $4 \times 13 \times 6$ cm$^3$ | HV | 1 | [7] |
| Capillary electrophoresis | 0.2 nM Fluorescein | $250 | $15 \times 7 \times 24$ cm$^3$ | 121 V, HV | 1 | [8] |
| Microfluidics | 100 nM Rhodamine 6G | n.s. | ≈ $5 \times 5 \times 0.1$ cm$^3$ | n.s. | 1 | [9] |
| Microfluidics | 2 nM Fluorescein | ≈$1,000 | $3 \times 3 \times 1.1$ cm$^3$ | ≈ $1-10$ W | 1 | [10] |
| Immunoassays | $10^4$ photons/mm$^2$ | ≈$50 | ≈ $5 \times 3 \times 8$ cm$^3$ | 5 V, ≈ $1-10$ W | 4 | [11] |
| Cas13a on paper | 6.8 nM Fluorescein | $15 | $7.5 \times 4.5 \times 11.5$ cm$^3$ | 5 V, < 1 W | 1 | this work |

another Cas nuclease, Cas12a, which instead of RNA recognizes dsDNA and develops a collateral ssDNA activity. This was utilized in a similar fashion to develop detection assays termed DETECTR (DNA endonuclease-targeted CRISPR trans reporter) [23] and HOLMES (one-HOur Low-cost Multipurpose highly Efficient System) [24]. This assay was later employed in combination with a facile sample preparation protocol termed HUDSON (Heating Unextracted Diagnostic Samples to Obliterate Nucleases) to detect viral RNA from clinical samples of Zika and Dengue patients [25]. To facilitate multiplexing and to provide a colorimetric readout, SHERLOCK 2.0 was later combined with a lateral flow assay [4].

Here we present a pocket-sized fluorescence detector optimized for measuring biochemical assays freeze-dried on paper in point-of-care settings. The detector consists of an assay cartridge and a detection unit (Fig 1). It can reliably measure fluorescein concentrations in the range of 10 to 1000 nM and was constructed from components costing altogether less than $15. The sample holder allows to sandwich a filter paper onto which the sample was placed between excitation and emission filter foils. The detection unit consists of the corresponding LED for fluorophore excitation and a light dependent resistor for emitted fluorescence detection, as well as auxiliary electronics. Detector calibration and automated time lapse measurements are controlled through a custom software. We developed a detailed calibration procedure including an estimation of measurement uncertainties to achieve nanomolar fluorophore detection sensitivity, a performance that is required for obtaining consistent data and time traces for common biochemical assays and that is comparable to what can be achieved with typically much more expensive laboratory equipment. The portability and low cost of our detector renders it useful in point-of-care applications.

In two application examples, we validate that our assay/detector system is suitable for a range of biological or chemical fluorescence assays. First, we quantified transcription of the

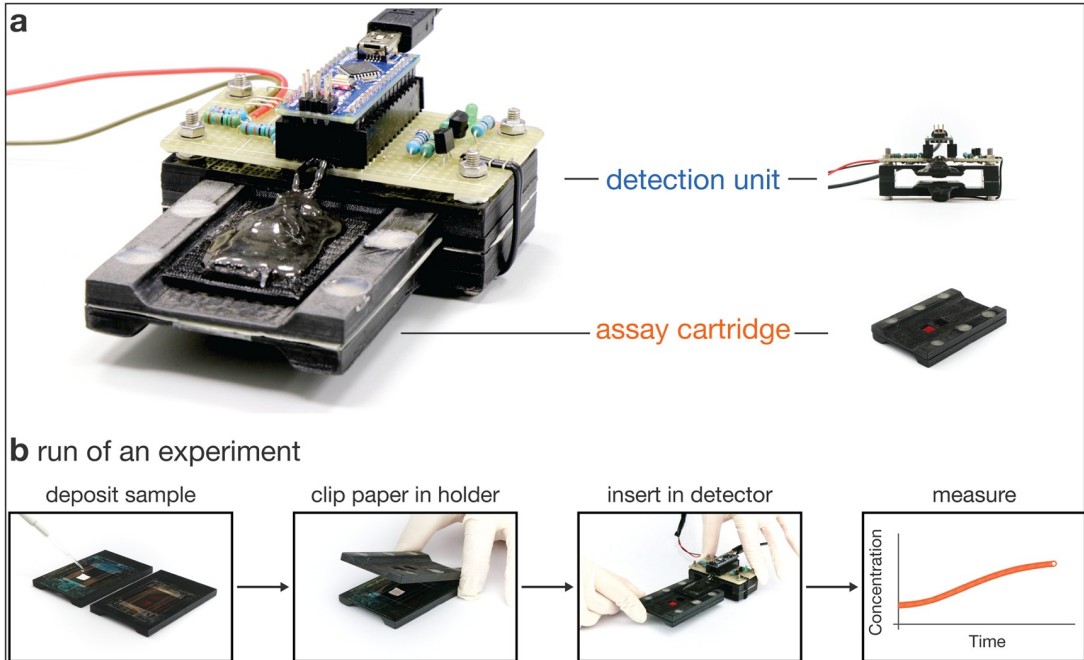

**Fig 1. Portable low cost detector.** (a) Photograph of the fully assembled detector (left) comprised of a detection unit and an assay cartridge (right). (b) Schematic of the experimental work flow. After sample deposition onto filter paper, the sample cartridge is clipped together and inserted into the detection unit. An automated time lapse measurement is then performed by the accompanying software.

fluorogenic RNA aptamer iSpinach, which becomes highly fluorescent in the presence of its ligand DFHBI *in vitro* [26, 27]. Second, nanomolar concentrations (above $\approx 4$ nM) of a target RNA were detected using a CRISPR/Cas13a paper based nucleic acid assay. As a proof-of-concept, the Cas13a detection system was developed for pathogen detection by undergraduate students as part of the 2017 iGEM Munich project, which aimed at rapid discrimination of bacterial from viral infections.

## Results

The design of our detector is based on the premise that fluorescence produced by biochemical reactions on filter paper should be detected in a most economic (cost as well as power consumption), yet reliable way. The main challenge in building a sensitive detector is maximizing the signal-to-background ratio. To ensure an optimal signal, state of the art laboratory equipment typically uses high-power light sources, focusing optics, and photo multipliers. Background signals are minimized through monochromators, filters, dichroic mirrors and by performing measurements at a 90˚ or 180˚ angle relative to the illumination source.

Previous solutions for low-cost detectors avoid some of these requirements, by making use of LEDs for excitation, in combination with photodiodes for detection [7, 8, 10]. However, common 90˚ or 180˚ excitation/emission geometries require appropriate optical components, compromising either sensitivity [7], or cost [8, 10]. They are also more difficult to combine with paper-detection formats.

We hypothesized that measuring at a 0˚ could allow for a high signal-to-background ratio, while omitting all optical components, except light filters. Since such an arrangement is placing the LED excitation light source and the LDR fluorescence sensor as close as possible (Fig 2a and 2b). In this setting, the choice of light filters is crucial to not compromise sensitivity, as bleed-through of excitation light to the sensor is the major source of background signal. We thus experimented with cheap photographic light filter foils and found a combination that works sufficiently well in our context. These foils are comparably thin, allowing to reduce the optical path to about 2 mm in total. Conceptually related approaches have been described for using polarizers [9] and low-cost bandpass filters [11], respectively.

Our final design features two subunits: a detection unit (Fig 2a, S1 Fig) and an assay cartridge (Fig 2b, S2 Fig). The assay cartridge sandwiches a glass fiber paper strip containing the sensor reaction mix between a set of color filter foils. After assembly, the cartridge is inserted into the detection unit (Fig 2c) that provides a blue LED as an excitation light source and a LDR as a sensor. The measurements are performed by a microcontroller (Arduino Nano) via a simple electronic circuit (S3 Fig) operated from a Windows laptop or tablet (Fig 2d), via a USB port providing 5 V, <1 W power.

### Operation of the detector

An overview of the typical operation workflow is given in Fig 1b. About 30 $\mu$l sample are pipetted onto the passivated filter paper and placed on the cartridge in front of the detection window. The cartridge is closed and inserted into the detector. After initiating the fluorescence reading via the operating software the user can inspect the measured data in a real-time plot. In this work, we have used either a laptop computer, or a Windows tablet (S5 Fig). Operating the detector on Android or iOS devices requires suitable software ports.

### Assay cartridge

Realization of a sensitive fluorescence detector requires maximization of the transmitted emission light reaching the sensor while simultaneously minimizing background signals [12]. The

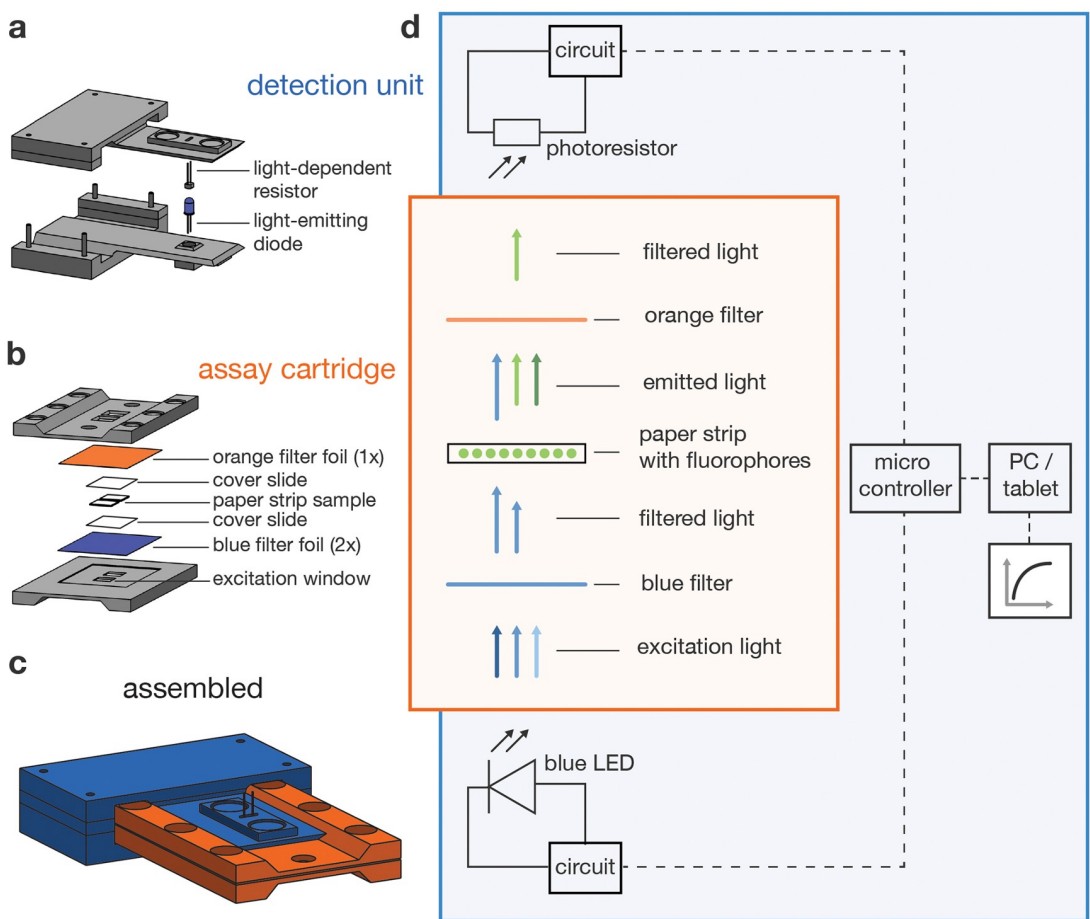

**Fig 2. Detector operating principle.** (a) The detector body includes the electronics and two flexible levers that extend to the LED and LDR. (b) The assay cartridge is made from two identical sides, each covered with lighting filter foils between which the filter paper containing the sample is placed, protected by two cover slides. (c) Sketch of the assembled detector. (d) Schematic illustrating the light path and electronic modules of the detector.

assay cartridge was designed accordingly, and we attempted to optimize the signal-to-background ratio at a minimum budget. The cartridge contains two filter paper sample spots and is built from two identical parts, each holding a filter foil transmission window for excitation and emission light, respectively (Fig 2b, S2 Fig). The flat cartridge design allows light source and sensor to be placed into direct proximity of the sample. The sensor can thus collect the maximum amount of emission light without using additional optical components such as lenses. In order to fix the sample and to exclude external background light, the cartridge is pressed together by six neodymium magnets.

A drawback of this design is that the sensor is placed in the direct light path of the source. Accordingly, appropriate optical filters are required to block background generated from directly transmitted excitation light, while permitting emitted light to pass through. As an alternative to expensive scientific grade optical filters, we used commercial filter foils for photographic lighting applications. To protect the filter foils from contamination, we covered them with microscopy cover slides for facile cleaning with ethanol and water between measurements. To identify an optimal excitation/emission pair, we purchased a sample block (LEE filters) and measured the filter spectra with a UV-Vis spectrometer. For the detection of fluorophores

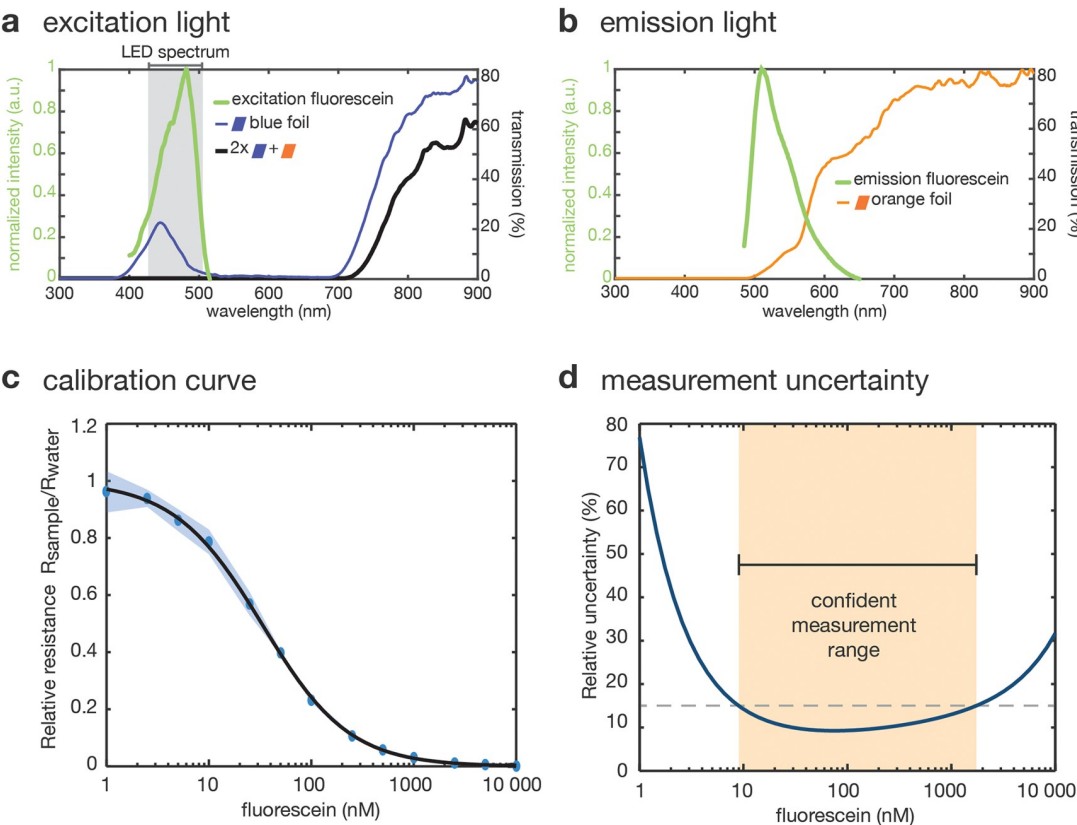

**Fig 3. Plastic filter foils for green fluorescence detection and calibration procedure.** (a) Excitation and (b) emission spectra of fluorescein (green) overlaid with the corresponding filter foil transmission spectra. The grey area in a) indicates the interval spanning 95% of the LED intensity. The combination of two blue and one orange filter foils allows for sufficient transmission of the excitation and emission light respectively, while blocking nearly all bleed-through excitation. (c) Calibration of the measured relative resistance for different dilutions of fluorescein over 3 replicates and 2 measurements per replicate. (d) Relative measurement uncertainty calculated as a function of the fluorescein concentration. The confident measurement range is taken as the range where the relative uncertainty is below 15%.

emitting in the green, a combination of the color "TOKYO BLUE" (LEE filters 071), as an excitation light filter, and "RUST" (LEE filters 777), as an emission light filter, represented the best trade-off. This combination blocks nearly all light up to 700 nm, while not limiting emitted light transmission too much (Fig 3). The detector can similarly be adapted for the measurement of other fluorophores by choosing alternative filter pairs and an appropriate excitation LED.

An additional source of background light is auto-fluorescence of the sample itself and in particular of the paper strip carriers used in our detector system. Filter paper was previously shown to be an excellent matrix for long-term storage of lyophilized reaction mixtures [16]. However, biological paper matrices composed of cellulose exhibit strong auto-fluorescence. We therefore chose glass fiber-based filter paper for precise and accurate fluorescence measurements with low auto-fluorescence [20]. Depending on the application, the filter paper can be passivated to avoid denaturation of sensitive components of the reaction mix, as detailed in the methods section.

## Detector unit

The assay cartridge described in the previous section fits into a dedicated slot of the detection unit. Both LED and LDR are mounted on two opposing levers that snap into the cavities of the

sample windows upon insertion of the cartridge (Fig 1b, S1 Fig). Magnets press both levers together to ensure stable measurements and to shield background light. The excitation LED (466 nm, 12 000 mcd $\equiv$ 70 mW/m$^2$ at 50 cm distance) is controlled by a microcontroller (Arduino Nano) via a transistor (S3 Fig).

For fluorescence light measurements, we use a cadmium sulfide (CdS) LDR, which is a very cost-effective light sensor. CdS LDRs have a maximum relative response for wavelengths around 520 nm and are therefore well-suited for sensing green fluorescence. Quantitative measurements using LDRs require careful consideration of their electric properties and associated measurement uncertainties. We therefore established a calibration procedure (S1 Appendix) and derived an analytical expression that allowed us to estimate the systematic measurement uncertainty as a function of the sample concentration (S1 Appendix) to determine the limit of detection (LOD) and the lower detection limit (LDL) of our setup.

**Light dependent resistor.** A typical response curve of an LDR follows the relation

$$R = I^{-\gamma} \ , \tag{1}$$

where $R$ is the resistance of the LDR, which changes with the light intensity $I$, and $\gamma$ is a characteristic parameter of the LDR, which can differ even between LDRs of the same type designation [28]. $R$ is measured via a voltage divider (S3 Fig) connected to an analog input pin of the micro controller (with an input impedance of 100 M$\Omega$ [29] (p.257)) to measure the voltage $U_{LDR}$ (0 to 5 V) in integer units from 0 to 1023. $R$ can then be computed as

$$R = \frac{R_{ref}}{\frac{U_0}{U_{LDR}} - 1}, \tag{2}$$

where the supply voltage $U_0$ is 5 V (or 1023). The reference resistance $R_{ref}$ was chosen as 750 k$\Omega$ corresponding to half of the LDR's maximum dark resistance (1.6 M$\Omega$), to ensure a maximum dynamic range of the measurement.

Further, we considered that LDRs respond slowly to changes in light intensity (within approximately 10 seconds [28]) and the resistance of an LDR is affected by Johnson-Nyquist noise. To compensate for this, the detector software turns on the excitation LED 30 seconds prior to the measurement, which was sufficient to equilibrate the LDR. Then, one data point is obtained by measuring $U_{LDR}$ 50 times in 50 ms intervals, which reduced the relative statistical uncertainty $\frac{\delta U_{LDR}}{U_{LDR}}$ to < 0.2% (S1 Appendix). The exact measurement protocol is defined through the detector control software.

**Calibration of the sensor.** Next, we derived a calibration function relating the sample fluorophore concentration $c$ with the measured resistance $R(c)$ (S1 Appendix):

$$\frac{R(c)}{R_b} = (1 + kc)^{-\gamma}, \tag{3}$$

where the calibration parameters are $\gamma$, the characteristic constant of the LDR, and $k$, which is a constant that depends on the spectral properties and light scattering effects of LED, filter foils, fluorophore, and the LDR, but not on the intensity of the excitation light. $R_b$ is the resistance measured for a blank sample, which was usually water as wet samples had a different background signal than dry ones, likely due to different light scattering from the filter paper.

To determine $k$ and $\gamma$, we obtained $R(c)$ and the corresponding standard error of the mean $\delta R(c)$ from a triplicate of dilution series ranging from 1 nM to 10 $\mu$M of fluorescein sodium salt (Fig 3c). Each series was preceded and ended with a blank measurement to confirm the absence of contaminations. To ensure that the LDR had equilibrated properly, each individual measurement was performed twice in direct succession. We then computed the relative

resistance $\frac{R(c)}{R_b}$ by dividing by the average $R_b = 1650 \pm 22$ kΩ obtained from all blank measurements ($N = 18$) and the uncertainty $\delta\frac{R(c)}{R_b}$ using Gaussian error propagation. Using the inverse square of the uncertainty as weights, we obtained $k = (0.0283 \pm 0.0010)$ nM$^{-1}$ and $\gamma = 1.054 \pm 0.019$, where $\delta\gamma$ and $\delta k$ are the asymptotic standard errors of the fit. Besides $\delta R_b$, $\delta\gamma$ and $\delta k$, we considered the uncertainties $\delta U_{LDR}$ and $\delta U_0$, which correspond to the 1 digit accuracy of measuring $U_{LDR}$ and $U_0$ with the microcontroller. By propagating these uncertainties through Eqs (2) and (3), we can compute the relative systematic uncertainty of a concentration measurement $\frac{\delta c}{c}$ as a function of concentration $c$ itself (S1 Appendix, Eq (S29), Fig 3d). The confident measurement range can then be defined as the interval where $\frac{\delta c}{c}$ is less than 15%, which is the case between 9 nM and 1730 nM. This is the range of fluorophore concentrations that can be reliably quantified, independently of the assay carried on the filter paper. The measurement range can be adjusted by choosing a different $R_{ref}$.

An alternative approach to determine the lower detection limit LOD (= LDL) and the lower quantification limit LOQ (also LQL) has been defined by the IUPAC [30]. Here, the LOD is simply the mean blank value plus 3 standard deviations $\sigma$ of the blank measurements. Because resistance is inversely related to $c$, this gives the following result:

$$LOD \quad = R_b + 3\sigma_{R_b} = 1372 \text{ kΩ} \cong 6.8 \text{ nM}$$
$$LOQ \quad = R_b + 10\sigma_{R_b} = 717 \text{ kΩ} \cong 42 \text{ nM}$$

## Detector operating software and GUI

The detector is complemented by a software user interface that can be operated from an external mobile device connected to the microcontroller's USB port. Setup parameters can be adjusted and saved in the user interface. Adjustable parameters include the calibration parameters, $R_b$, $k$ and $\gamma$ together with their uncertainties, the reference resistance $R_{ref}$, the number of averaged measurements per data point, the equilibration time of the LDR, the number of data points to be collected, and the length of the time interval between the collected data points (Fig 4b). The detector calibration interface facilitates the measurement series acquisition for computing the calibration parameters (Fig 4a). Once calibrated, the detector can perform automated time-series measurements during which the data, including measurement uncertainties, are plotted and saved in real time (Fig 4c).

## Example applications

**Monitoring *in vitro* transcription reactions.** In a first application example that allowed us to validate our detector architecture, we measured fluorescence time traces obtained during *in vitro* transcription of the fluorogenic iSpinach-RNA aptamer [27]. The corresponding sample contained T7 RNA polymerase, template DNA coding for iSpinach under a T7 RNAP promoter and the iSpinach fluorophore DFHBI (Fig 5a). In this reaction, the fluorescence signal increases proportionally with the formation of iSpinach:DFHBI complexes, which directly follows iSpinach RNA synthesis. The time traces obtained for different template concentrations are shown in Fig 5b, along with reference experiments conducted in a plate reader (inset).

**CRISPR/Cas13a-based RNA detection.** We then applied the detector for nucleic acid detection using the CRISPR/Cas13a-based platform SHERLOCK [20], whose molecular reaction mechanism is sketched in Fig 6a. Cas13a is an RNA-dependent RNase that can be programmed to bind an ≈20 nucleotide (nt) long target RNA sequence with high specificity by designing a crRNA (CRISPR-RNA) complementary to its target sequence. Upon binding and cleaving of a target, Cas13a develops an unspecific collateral RNase activity. This collateral

**a** user interface

**b** control parameters

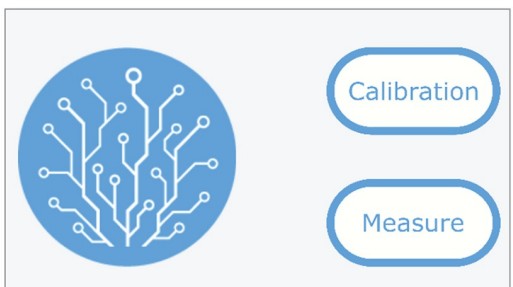
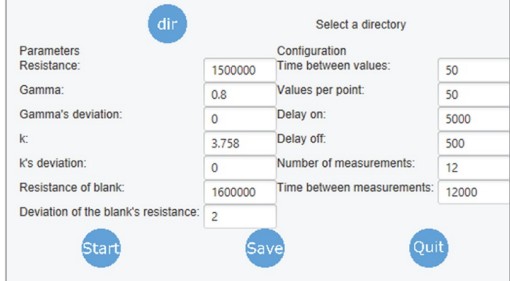

**c** measurement

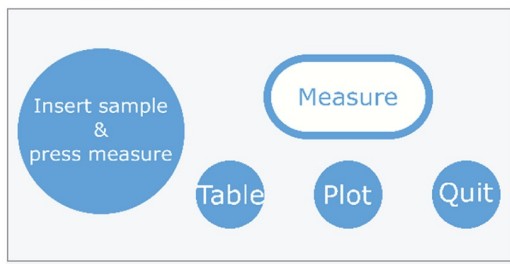
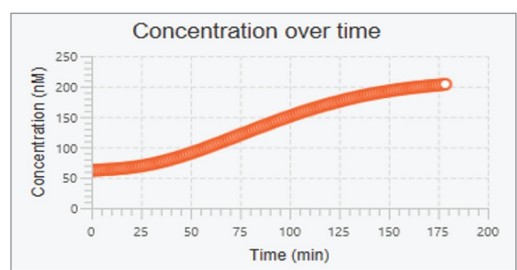

**Fig 4. Detector software.** (a) The user interface of the software allows to perform a calibration (to determine the dependence of the measured resistance on fluorophore concentration) or directly measure a sample. (b) Several parameters can be adjusted, such as the calibration parameters and the frequency of data acquisition. c) During measurements, acquired data can be plotted in real time.

**a** spinach transcription

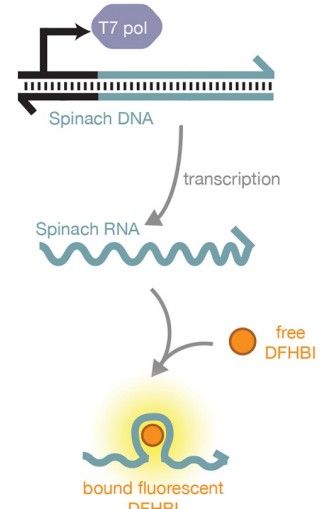

**b**

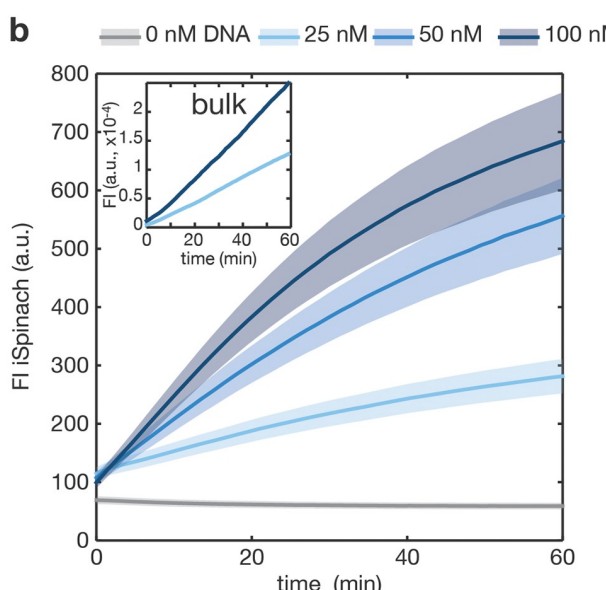

**Fig 5. Detection of the transcription of a fluorescent aptamer.** (a) Scheme of the transcription of iSpinach aptamer, which binds the fluorophore DFHBI and increases its fluorescence. (b) Measurement of the concentration of iSpinach:DFHBI complex during transcription on the detector and in bulk (inset), with various concentrations of DNA template: 0 nM (grey), 25 nM (light blue), 50 nM (median blue) and 100 nM (dark blue). Thick line and shaded area represent respectively mean and measurement uncertainty as computed in S1 Appendix.

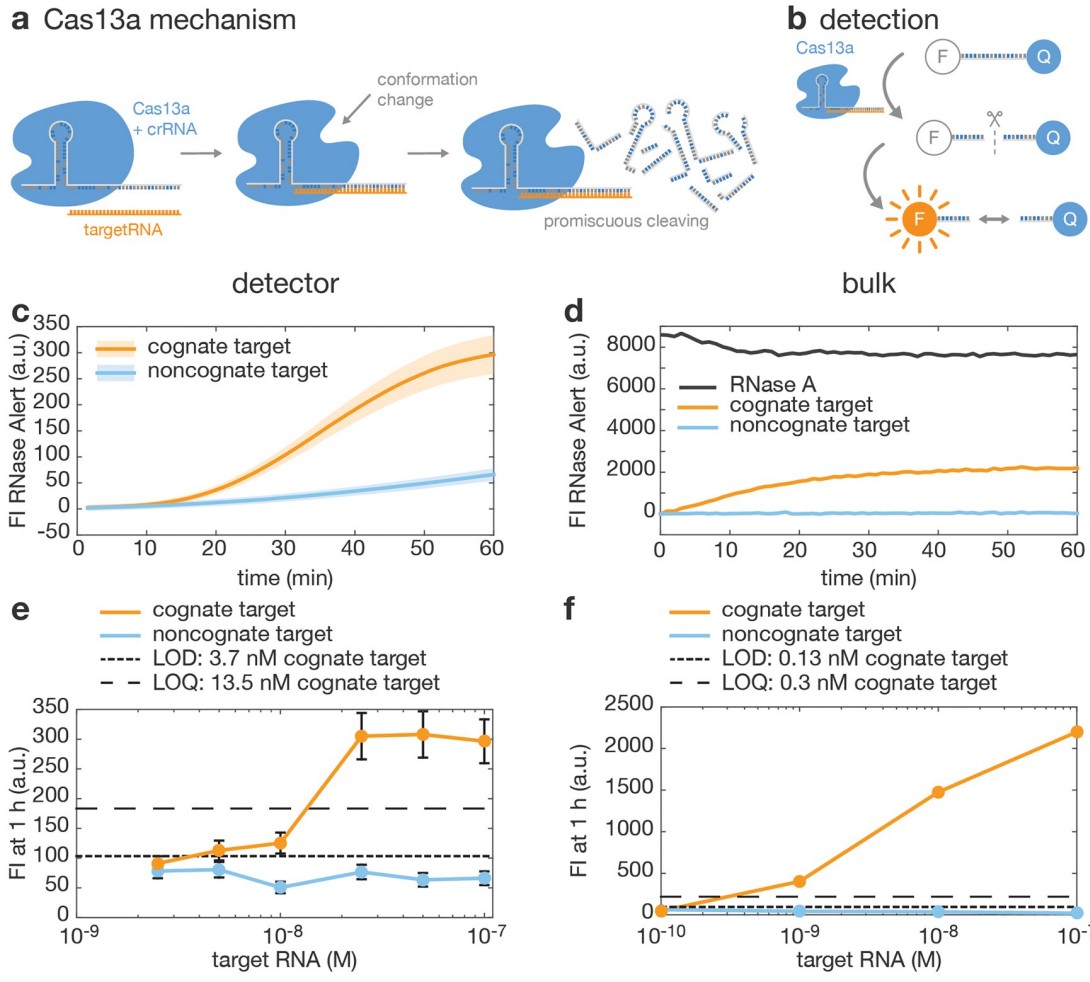

**Fig 6. Detection of Cas13a activity.** (a) Mechanism of action of Cas13a: the protein (blue) forms a complex with a crRNA (grey) that consists of a Cas13a-handle and a sequence complementary to the target RNA (orange). Upon binding of the target RNA to the Cas13a-crRNA complex, Cas13a undergoes a conformational change that activates promiscuous RNase activity: Cas13a becomes an unspecific RNase [31]. (b) Mechanism of detection of Cas13a activity: a short RNA strand is modified with a fluorophore and a quencher. Upon cleavage by Cas13a, the fluorophore is released from the proximity of the quencher, and therefore fluorescence increases. (c-d) Measurement of Cas13a activity with 100 nM target RNA on paper in the detector (c) or in bulk using a plate reader (d). Activity in presence of a cognate (*i.e.* complementary to the crRNA) RNA target is compared to a non-cognate target. Residual activity in the presence of non-cognate target is likely due to an unspecific activity of Cas13a. In (d), positive control in bulk contains RNase A. (e-f) Response of the assay to increasing concentrations of cognate or non-cognate RNA target, in the detector (e) or in bulk (f). The LOD and LOQ as calculated based on the mean and standard deviation of background measurements are shown with dotted lines. Thick line and shaded area (c) and dots and error bars (e) represent mean and measurement uncertainty as computed in S1 Appendix.

activity can be monitored with the self-quenching fluorescent RNA beacon, RNaseAlert (Fig 6b). In combination with an isothermal amplification step (RPA-TX), this approach allows for highly specific and sensitive nucleic acid detection. As it was previously shown that all necessary reagents can be lyophilized and stored on filter paper, our detector could potentially be used to apply SHERLOCK in low resource settings. We tested the feasibility of assessing Cas13a activity with our detector by assembling the reaction on paper with a cognate target RNA complementary to a crRNA and a non-cognate target that did not hybridize to the crRNA. An exemplary fluorescence time trace showing the target specific cleavage of RNaseA-lert by Cas13a, as well as the control experiment are shown in Fig 6c. The 30 µL sample

contained 20 nM Cas13a, 150 nM crRNA, 100 nM target RNA and 200 nM RNaseAlert on BSA-passivated filter paper. A fast increase in fluorescence signal upon recognition of the cognate target RNA makes this sample distinguishable within 20 minutes from background activity with a non-cognate RNA target. This fast and specific detection of a nucleic acid sequence makes the combined SHERLOCK method and our fluorescence detector suitable for POC applications.

Identical experiments were conducted in a 384-well plate and measured with a commercial plate-reader, as shown in Fig 6d. Comparing the experiments in bulk and on paper, the curves are qualitatively similar and show comparable signal amplification. However, the time required to reach the maximum fluorescence in the detector was approximately three times (60 minutes) the time required in the plate reader (20 minutes). This suggests an overall lower catalytic activity in the filter paper assay, which may be due to inactivation or loss of one or a combination of the reaction components. Similarly, the LOD and LOQ are lower in bulk measurements compared to measurements in the detector, although they remain in both cases in the sub-nanomolar to nanomolar range, which is consistent with literature values [20].

## Conclusion

We developed a portable low-cost fluorescence detector and successfully used it to monitor biochemical reactions on disposable paper strips. To achieve sensitive and reproducible measurements with low cost components such as an LDR and photography filter foils, we established a thorough calibration procedure and carefully shielded background light. This allows for straight-forward adaptation of fluorescence-based diagnostic tests from a laboratory-based to an infield application, without the need to engineer an alternative visual readout system.

As an exemplary application, we monitored Cas13a activity on passivated, non-auto-fluorescent filter paper, which can be used for nucleic acid testing, for instance detecting viral or bacterial infections. The use of filter paper enables the storage and distribution of the biochemical reaction components, which is critical for a test to be employed in a point-of-care scenario. An in-field application might benefit from multiplexing capabilities, which would allow the simultaneous diagnosis of multiple patients or multiple diseases and include the appropriate controls. Besides upgrading the detector design with multi-channel and/or multi-sample functionalities, multiplexing capabilities could also be engineered into the detection biochemistry. In the case of our Cas13a-based assay this could be achieved by implementing computational modules, for instance using strand displacement guide RNAs [32].

CRISPR/Cas systems and cell-free protein synthesis are tools that are simple to design and operate and have therefore recently been popularized as an educational kit termed BioBits™, which uses a qualitative fluorescence imaging chamber [33, 34]. The flexibility, low cost and facile assembly of our detector-cartridge system renders it suitable for quantitative measurements in such teaching activities. Furthermore, in combining hardware, software, physics and biochemistry concepts, interdisciplinary teaching activities can be developed for this platform in the future.

## Materials and methods

### Expression of Cas13a

Cas13a was expressed from p2CT-His-MBP-Lbu_C2c2_WT, which was obtained from the Jennifer Doudna lab via Addgene (Addgene plasmid No. 83482; RRID:Addgene_83482; http://n2t.net/addgene:83482) [22]. The His-MBP tagged Cas13a plasmid was transformed into Rosetta *E. coli*. LB-carbenicillin plates were plated with a glycerol stock of the cells (50% glycerol, 50% saturated culture) and incubated at 37˚C overnight. Single colonies were picked and

resuspended in a preculture of 50 mL LB-medium supplemented with 100 $\mu$g/ml of the antibiotic carbenicillin. 10 mL of the preculture was further grown in 1 L 2x YT medium (Carl Roth) with 100 $\mu$g/ml carbenicillin. Cells were grown aerobically in a shaker at 37˚C up to an optical density of 0.6-0.8, before inducing Cas13a production with 1 mM IPTG in an overnight culture at 16˚C. Cells were pelleted the next day by centrifugation at 5000 rcf for 30 minutes at 4˚C. The supernatant except 50 mL was discarded. The cell pellet was resuspended in the remaining 50 mL and centrifuged again at 4500 rcf for 10 min at 4˚C. After discarding the supernatant, pellets were flash frozen with liquid nitrogen and stored until further use at -80˚C.

## Purification of Cas13a

Cell pellets were thawed and resuspended in 20 ml lysis buffer (50 mM Tris pH 7, 500 mM NaCl, 5% glycerol and 1 mM TCEP). One protease inhibitor tab (cOmplete, Roche) was added and cells were sonicated at 50% amplitude, 20 s pulse, and 10 s pause (Sonopuls mini20, Bandelin). Lysed cells were centrifuged at 6000 rcf for 30 min. A volume of Ni-NTA Agarose beads (Qiagen, Germany) equivalent to 1/4 of the volume of cell supernatant was centrifuged down for 15 s on a table-top centrifuge. After discarding the supernatant the bead pellet was resuspended in lysis buffer (1/2 the volume of the initial cell supernatant). These beads suspended in lysis buffer were centrifuged again and the supernatant was discarded. Then cell supernatant was dispersed into the washed beads and incubated for 60 min at 4˚C while shaking to allow the proteins to bind the beads. This mixture was loaded unto a spin-column, which was subsequently washed twice with 2.5 mL wash buffer (50 mM Tris pH 7, 500 mM NaCl, 5% glycerol, 25 mM Imidazole and 1 mM TCEP). The protein was then eluted 3 times with 0.5 mL elution buffer each (50 mM Tris pH 7, 500 mM NaCl, 5% glycerol, 250 mM Imidazole and 1 mM TCEP). Lysate, flow-through, wash and elution fractions were collected individually and analyzed on a 10% SDS gel (S4a Fig). Protein-containing fractions were dialyzed overnight at 4˚C in a gel filtration buffer (20 mM Tris pH 7, 200 mM NaCl, 5% Glycerol and 1 mM TCEP) along with TEV protease (purified with a similar protocol and Ni-NTA Agarose beads from *E. coli* BL21 star (DE3) cells transformed with pSB1C3-His-BBa-K1639008) at a ratio of 1 protease to 100 MBP-Cas13a to cleave off the His-MBP tag. The following day protease treated protein was purified on a Ni-NTA column to remove the His-MBP tag as described above, except that this time the flow through was collected and analyzed on an SDS gel with subsequent Bradford protein concentration determination (S4b Fig). Purified Cas13a was further concentrated using spin-concentrators (Amicon Ultra 15 ml centrifugal tubes) to about 500 $\mu$l to 1 ml final volume for S200 size exclusion column purification (Äkta). Cas13 concentration after size exclusion was about 697 $\mu$g/ml (5 $\mu$M). All eluted fractions from the SEC peak were characterized on a 10% SDS gel and protein aliquots were flash frozen in liquid nitrogen and stored at -80˚C.

## Passivation of filter paper

Filter paper (Whatman Grade 934-AH glass microfiber filters) was cut into $\approx 1 \times 1$ cm$^2$ squares and placed in a glass Petri dish using tweezers, both of which were cleaned with RNase ZAP (Sigma-Aldrich, Germany). Then, 5% w/v RNase free BSA solution (VWR, No. 0332-25G) was poured over the paper. The Petri dish was covered with aluminum foil and incubated for 4 h. The papers were then taken out and, to minimize sticking, placed upright against the walls of a second, identically treated petri dish and allowed to dry for 1 h. Finally, residual moisture was removed by baking the BSA-treated paper sections in an oven at 100˚C for 30 min.

### *In vitro* transcription

The *in vitro* transcription reaction contained 500 nM T7 RNA polymerase (kindly provided by Dr. Sandra Sagredo), with 3, 6, or 12 nM iSpinach DNA (IDT, USA, Ultramers) template in transcription buffer (1x RNAPol Reaction Buffer (NEB, No. B9012, 40 mM Tris-HCl, 6 mM MgCl$_2$, 1 mM DTT, 2 mM spermidine, pH 7.9 at 25˚C), supplemented with 12 mM MgCl$_2$, 25 mM KCl, 40 $\mu$M DFHBI-1T (Lucerna), 4 mM rNTPs each, and 1 U/$\mu$l RNase Inhibitor, Murine (NEB, No. M0314)).

Purified guide crRNA and template RNA was obtained by following this *in vitro* transcription as a 100 $\mu$l batch reaction with a DNA digestion step (0.035 U/$\mu$l DNaseI, 1x DNaseI Buffer, NEB No. M0303S, 37˚C, 1h) and a phenol-chloroform extraction protocol in a 5'-Phase Lock Gel Heavy (VWR, USA). The RNA was precipitated with ethanol and resuspended in nuclease-free water. RNA concentration was measured by denaturing PAGE. The gel consists of 8 M urea (Carl Roth, Germany, No. 2317), 1x TBE Buffer (Carl Roth, No. 3061), 15% (v/v) Acrylamide 29:1 (Carl Roth, No. A121), 0.1% (v/v) TEMED (Carl Roth, No. 2367), 0.1% (w/v) APS (Sigma-Aldrich, Germany, No. A3678), in double-distilled H2O (total volume 10 mL). The gel was cast in cassettes (Bolt Empty Mini, Thermo Fisher Scientific, USA), and the running buffer is 1x TBE. The gel was pre-run for 30 min at 12.5 V/cm with a running temperature of 40˚C. RNA samples were mixed with 2x RNA loading dye (Thermo Fisher Scientific, USA, No. R0641), denatured at 95˚C for 5 min and directly put on ice to prevent the RNA from refolding. RNA samples were then loaded onto the gel and ran at 12.5 V/cm for 1 h 15 min at 40˚C. The resulting gel was stained with 1x SybR Green II (Thermo Fisher Scientific, No. S7586). Then a fluorescence image of the gel was acquired and the intensity of the RNA band was quantified against an RNA ladder (low Range Riboruler, Thermo Fisher Scientific, No. SM1831) with ImageJ (S4c Fig).

### Cas13a assay

For the Cas13a RNA detection assays crRNA was first diluted with processing buffer (1x is 20 mM HEPES pH 6.8, 50 mM KCl, 5 mM MgCl$_2$ and 5% glycerol) to a final assay concentration of 150 nM crRNA in 1x processing buffer, and incubated for 5 min at 65˚C to ensure complete folding of the crRNA. Then, Cas13a was added to a final assay concentration of 20 nM and the sample was incubated for 10 min at 37˚C to mediate binding between Cas13a and crRNA. Finally, 200 nM of RNase Alert (Thermo Fischer Scientific), 1.5 U/$\mu$l of RNase Inhibitor, Murine (NEB, No. M0314) and 100 nM of target RNA (all final concentrations in the assay) were added on ice. The solution was mixed by pipetting up and down, and immediately applied to the filter paper and measured.

### Calibration of the detector

We first prepared a stock solution containing 1 mM fluorescein sodium salt in water. This was then diluted to 1, 2, 5, 10, 20 nM, etc. up to 10 $\mu$M. For one measurement, 30 $\mu$l of the sample was placed on a $1 \times 1$ cm$^2$ piece of filter paper and measured with the detector. The sequence of measurements for one dilution series as implemented in the software as an automated calibration routine was (blank)$_2$-(1 nM)$_2$-(2 nM)$_2$-(...)$_2$-(blank)$_2$, where subscript 2 indicates that each measurement is directly repeated to ensure that the LDR had equilibrated properly, and the blank measurements should confirm the absence of contaminations. After completing this procedure in triplicates, the resulting data was processed python script (S1 File) to compute $\frac{R}{R_b}$ and $R(c)$ by fitting $k$ and $\gamma$ with gnuplot.

## Time trace measurement with the detector

To initiate a time trace measurement, the desired number of data points and the time interval between measurements was specified in the software interface. Then, about 30 $\mu$l sample are pipetted onto the passivated filter paper and placed on the cartridge in front of the detection window (Fig 1b). The cartridge was closed, inserted into the detector. Then the measurement was started via the operating software and saved in real-time in a.csv format. Plate reader reference curves were recorded using a 384 wellplate (4titude, Ultravision) on a CLARIOstar 2016 plate reader at 470-15 nm excitation with a 491.1 dichroic filter, a 515-20 nm emission and a gain of 1500 by incubating 15 $\mu$l reaction volumes at 30˚C.

## Supporting information

**S1 Appendix. Derivation of calibration function and analysis of measurement uncertainties.**
(PDF)

**S1 Fig. Detection unit.** (a) CAD drawing, (b) side view photograph and (c) front view photograph illustrating the assembly of the detection unit. The 3D printed parts are first treated with sand paper and the holes for the screws are drilled. Magnets, LED and LDR are inserted and glued into the corresponding cavities. For the LED and LDR we mixed the glue with graphene to block transmission of background light. Then the circuit board carrying the microcontroller (S3 Fig) is assembled on top of the unit and fixed with screws. Finally, the LED and LDR are connected to the circuit board.
(PNG)

**S2 Fig. Assay cartridge.** (a) CAD drawing and (b) photograph illustrating the assembly of the assay cartridge. The 3D printed parts are first treated with sand paper. Then the magnets are glued into the corresponding cavities. The filter foils and protective cover slides are cut into the appropriate size, assembled in front of the detection window and fixed with Scotch tape. The filter foils are covered with microscope cover slides to facilitate cleaning with ethanol and water between measurements. To obtain a clean optical pathway, the transmission windows must be free of Scotch tape. A piece of filter paper carrying the sample is placed in front of the transmission window.
(TIF)

**S3 Fig. Circuit diagram and board layout of the detection unit.** (a) The blue excitation light LED and a green status LED are controlled by NPN transistors via the digital pin-out of the microcontroller. The resistance of the LDR changes according to the intensity of incoming emission light and is measured via a voltage divider using an analog input-pin of the microcontroller. (b) overlay, (c) top, and (d) bottom view of the used circuit board layout for soldering.
(PNG)

**S4 Fig. Cas13a and RNA preparation.** SDS-gels of Cas13a Ni-NTA purification after cell lysis (a) and after TEV protease digestion (b). (c) Gel-electrophoretic analysis of *In vitro* crRNA and targetRNA transcription for Cas13a assay.
(PNG)

**S5 Fig. Operation of the detector outside the laboratory.** The detector can be operated and powered from a Windows tablet.
(JPG)

**S1 File. Detector operating software and CAD files.** Are available via Github: https://github.com/Katzi93/Fluorescence_detector.
(ZIP)

**S2 File. Detector components.** Bill of materials.
(XLSX)

**S3 File. DNA sequences.** Lbu-Cas13a, iSpinach, crRNA and Cas13a.
(PDF)

# Acknowledgments

This research was conducted as part of the joint TUM-LMU iGEM Munich team 2017. We gratefully acknowledge Erika Chacin De Leonardis, Christoph Neumayer, Teeradon Phlaira-harn, Julian Reinhard, Florian Rothfischer, Robert Strasser, Patrick Wilke, Milica Zivanic, Benjamin Aleritsch, Dong-Jiunn Jeffery Truong, and Gil Gregor Westmeyer for their contributions during the iGEM competition phase of this project. We thank the GRK2062 'Molecular Principles of Synthetic Biology' for discussions and advice throughout the project.

# Author Contributions

**Conceptualization:** Florian Katzmeier, Lukas Aufinger, Aurore Dupin, Friedrich C. Simmel, Michael Heymann.

**Data curation:** Florian Katzmeier, Lukas Aufinger, Matthias Lenz, Ludwig Bauer, Sven Klumpe, Dawafuti Sherpa, Maximilian Honemann, Igor Styazhkin, Michael Heymann.

**Formal analysis:** Aurore Dupin.

**Investigation:** Florian Katzmeier, Matthias Lenz, Ludwig Bauer, Sven Klumpe, Dawafuti Sherpa, Benedikt Dürr, Maximilian Honemann, Igor Styazhkin.

**Resources:** Friedrich C. Simmel.

**Software:** Jorge Quintero.

**Supervision:** Lukas Aufinger, Aurore Dupin, Friedrich C. Simmel, Michael Heymann.

**Visualization:** Aurore Dupin, Benedikt Dürr, Michael Heymann.

**Writing – original draft:** Florian Katzmeier, Lukas Aufinger, Aurore Dupin, Friedrich C. Simmel, Michael Heymann.

**Writing – review & editing:** Jorge Quintero, Matthias Lenz, Ludwig Bauer, Sven Klumpe, Dawafuti Sherpa, Benedikt Dürr, Maximilian Honemann, Igor Styazhkin.

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
