## [Decision Letter · Decision Letter 0]

30 Jul 2019

PONE-D-19-18880

A low-cost fluorescence reader for in vitro transcription and nucleic acid detection with Cas13a

PLOS ONE

Dear Dr. Heymann,

Thank you for submitting your manuscript to PLOS ONE. After careful consideration, we feel that it has merit but does not fully meet PLOS ONE’s publication criteria as it currently stands. Therefore, we invite you to submit a revised version of the manuscript that addresses the points raised during the review process.

Please consider all the points of the reviewers, including a limit of detection assay and controls. Detecting a real sample is not absolutely required.

We would appreciate receiving your revised manuscript by Sep 13 2019 11:59PM. To enhance the reproducibility of your results, we recommend that if applicable you deposit your laboratory protocols in protocols.io, where a protocol can be assigned its own identifier (DOI) such that it can be cited independently in the future. For instructions see: http://journals.plos.org/plosone/s/submission-guidelines#loc-laboratory-protocols

We look forward to receiving your revised manuscript.

Kind regards,

Mark Isalan

Academic Editor

PLOS ONE

Journal Requirements:

[none].

Reviewers' comments:

Reviewer's Responses to Questions

**Comments to the Author**

1. Is the manuscript technically sound, and do the data support the conclusions?

Reviewer #1: Partly

Reviewer #2: Yes

Reviewer #3: Yes

2. Has the statistical analysis been performed appropriately and rigorously? 

Reviewer #1: Yes

Reviewer #2: Yes

Reviewer #3: I Don't Know

3. Have the authors made all data underlying the findings in their manuscript fully available?

Reviewer #1: Yes

Reviewer #2: Yes

Reviewer #3: Yes

4. Is the manuscript presented in an intelligible fashion and written in standard English?

Reviewer #1: No

Reviewer #2: Yes

Reviewer #3: Yes

5. Review Comments to the Author

Reviewer #1: Summary

This manuscript describes the development of a low-cost fluorescence spectrophotometer designed to facilitate fluorescence measurements at the point-of-care. The authors have presented a very clever and resourceful design for a fluorescence detector. Given the growing interest in point-of-care diagnostics, I think that this work will be of broad interest to the synthetic biology and biotechnology community. As such, I think this piece is well suited for PLoS One and recommend that this manuscript be published after a few major and minor revisions, explained in detail below.

Major revisions

The authors argue that their fluorescence detector can be used in low-resource settings, but it appears that operation of the detector requires software run on an external computer, which may not be available. Could the software instead be run on a smart phone or tablet? If so, I recommend including this as a demonstration in the manuscript to support the claim that the detector can be used in low-resource settings. Broadly, I ask that the authors somehow address this issue of portability and accessibility before claims about utility in low-resource settings can be made.

The authors argue in their abstract that DIY construction of their fluorescence detector could serve as an educational activity that combines electrical engineering, computer engineering, and biochemistry concepts. However, no framework or target audience (high school, undergraduate, etc.) for teaching such concepts is discussed. Please either provide example curriculum pieces (powerpoint, worksheets, etc.) to support this claim or remove it from the manuscript.

In the results section, I would have liked to read more about the rationale and strategy behind the development of the spectrophotometer. How did you design the device? Did you deconstruct high-tech equipment or model after existing low-cost fluorescence detection equipment? I’d imagine some of the choices made, such as the use of photography lighting filters instead of expensive scientific light filters, might not have been tried before and would be of interest to others seeking to replicate or expand upon this work. Since the design of the fluorescence spectrometer is the crux of this work, I recommend that the authors add discussion of their detector design strategy placing it in the context of previous efforts.

Minor revisions

A low-cost fluorescence imager designed to enable qualitative or semi-quantitative analysis of cell-free reactions as part of hands-on educational activities was recently described (https://advances.sciencemag.org/content/4/8/eaat5107). Please include this citation in the discussion when educational applications are discussed.

I noticed on the order of 10 typos throughout the manuscript - please address before resubmission.

Reviewer #2: This study (by Katzmeier, Aufinger, Dupen, Quinteiro et al.) develops a low-cost fluorescence reader, and demonstrate applications with Cas13 detection of RNA, and the RNA aptamer Spinach.

I think the technology described has promise, but would like to see additional validation to show the sensitivity of Cas13 detection using the device for readout, as well as a few additional controls to validate the performance. With these additions, the paper should be suitable for publication.

Major comments:

1. Does the cartridge need to be cleaned in between uses? It is unclear if any sample from the filter paper might be carried over between runs. The authors should clarify this point, as it influences the utility of the device.

2. The authors should perform a limit of detection assay using Cas13a and their fluorescent detector, if they want to use Cas13 as an application of their device. It is not sufficient to show a single experiment, with 100 nM target, and make claims in the abstract about detection in the "nanomolar range".

3. Similarly, Table 1 is misleading, as the fluorescein calibration experiments (shown in Fig. 4) appear to be independent of Cas13 - as far as I can tell no sensitivity analysis has been performed with Cas13 (just with fluorescein).

4. Figure 6b does not have any negative controls, please rerun this experiment with some negative controls (or show them in the plot).

Minor comments:

1. Some of the figures could be combined to improve clarity (e.g., 3 and 4; possibly 1 and 2)

2. What is the final concentration of Cas13a and crRNA in the detection reactions? It is unclear (as written) in the Methods section.

3. The fluorescence in the presence of noncognate target is increasing over time in panel 7c and 7d, to approximately 1/3 of the maximum signal detected for Cas13 in the presence of cognate target after 1 hour of detection. This is remarkably high. Can the authors comment on why this is the case? Is there some sort of impurity present in the Cas13 or crRNA? This effect does not appear to be specific to the detector system being used, so presumably is coming from one of the reagents.

4. There appears to be a broken reference (a "?" in line 72). Please fix, and proofread the manuscript to ensure that no other such typos are present.

5. The authors reference several point of care technologies, but have omitted a recent demonstration of Cas13 detection of viruses at the point of care, which was published alongside the SHERLOCK 2.0 paper: doi: 10.1126/science.aas8836

Reviewer #3: In this manuscript, the authors developed a low-cost fluorescence reader for POCT application. With the employment of the CRIPSR-Cas13 collateral cleavage activity against RNA, target RNA can be conveniently detected by the developed reader with high sensitivity. And this invent surely has the potential to be widely used in both clinical and household application scenarios. I have only several minor concerns for the authors to address before the work can be accepted for publication.

1) The filters can be polluted by the paper strip. How to avoid the pollution?

2) In figure 7, longer reaction time is required for the detector, and the authors proposed several possibilities. However, it is still highly recommended that the authors may test with increased amount of enzyme with the detector.

3) To demonstrate the practicability of the device and the system, the authors may need to detect a real clinical or nonclinical sample with the system.

4) Besides of Cas13, Cas12 has also been used for CRISPR-Dx (i.e. HOLMES and DETECTR), and the authors may need to discuss them or at least describe in the introduction part. If the reader is also compatible with the Cas12 system, more readers will be interested.

6. PLOS authors have the option to publish the peer review history of their article (what does this mean?). If published, this will include your full peer review and any attached files.

Reviewer #1: No

Reviewer #2: No

Reviewer #3: No

---

## [Author Response · Author response to Decision Letter 0]

7 Oct 2019

Reviewer #1: 

Summary: This manuscript describes the development of a low-cost fluorescence spectrophotometer designed to facilitate fluorescence measurements at the point-of-care. The authors have presented a very clever and resourceful design for a fluorescence detector. Given the growing interest in point-of-care diagnostics, I think that this work will be of broad interest to the synthetic biology and biotechnology community. As such, I think this piece is well suited for PLoS One and recommend that this manuscript be published after a few major and minor revisions, explained in detail below.

Major revisions

The authors argue that their fluorescence detector can be used in low-resource settings, but it appears that operation of the detector requires software run on an external computer, which may not be available. Could the software instead be run on a smart phone or tablet? If so, I recommend including this as a demonstration in the manuscript to support the claim that the detector can be used in low-resource settings. Broadly, I ask that the authors somehow address this issue of portability and accessibility before claims about utility in low-resource settings can be made.

The reviewer questioned that a laptop computer may not be available, require an AC power outlet, or be inconveniently large in low-resource settings. While various compact low-priced models are available, we now also demonstrate detector compatibility with low-power devices. For this, we operated the detector from a Windows tablet without major modifications, and updated Figure 2, Fig S5, and the text accordingly. 

Line 151:

“In this work, we have used either a laptop computer, or a Windows tabled (Fig S5). Operating the detector on Android of iOS devices requires suitable software ports.”

Although we do not demonstrate the operation of the detector with iOS or Android devices, adapting the software accordingly should to our understanding be possible for an expert in this field. While this is beyond our own expertise, we chose to provide all source code as open source in the hope that such portations will be performed in the future.

Regarding the power requirements, smartphones have enough battery capacity to theoretically operate the detector for about ~4000 mAh*5 V/1 W = 20 h. However, the USB/ lightning ports may be limited in their output power to ~0.5-2.5 W in which case it might be necessary to use an additional power bank.

The authors argue in their abstract that DIY construction of their fluorescence detector could serve as an educational activity that combines electrical engineering, computer engineering, and biochemistry concepts. However, no framework or target audience (high school, undergraduate, etc.) for teaching such concepts is discussed. Please either provide example curriculum pieces (powerpoint, worksheets, etc.) to support this claim or remove it from the manuscript.

All engineering and biochemistry work presented in the manuscript was completed by bachelor and master level students during their iGEM summer project as part of their scientific university education. The team comprised of students from diverse degrees, ranging from biotechnology to physics and computer science. While participation in the iGEM project was recognized with credit points that count towards the respective degree requirements, we did not perform conventional lab exercises with worksheets and a condensed work program. Instead students were tasked to explore primary literature and to design and conduct experiments by themselves with supervisors available for counseling. 

The genuinely positive feedback about the cross-disciplinary learning experience that we supervisors received from participating students prompted us to comment on the potential utility of the presented detector and auxiliary assays in educational settings in both the abstract and conclusion sections. 

Since it was not our intention, we apologize to the reviewer for having invoked the notion to expect a ready-made teaching lab tool kit. This in fact is well beyond the scope of our current work. In light of the general custom to also comment on suitable future directions and developments we like to comment on possible future applications in science teaching. Especially the many cultural and organizational differences in the international education systems motivated us to provide full open source access to all aspects of the detector including the detailed technical supplement to help interested to implement the system for their specific needs – including in education.

We thus now write: 

Abstract, line 33: 

“Furthermore, our open-source device may be used in educational settings, through providing low cost instrumentation for quantitative assays or as a platform to integrate hardware, software and biochemistry concepts in the future.”

Conclusions, Line 325: 

“CRISPR/Cas systems and cell-free protein synthesis have become popular tools that are simple to design and operate and have therefore recently been popularized as an educational kit termed BioBits, including a qualitative fluorescence imaging chamber [29, 30]. The flexibility, low cost and facile assembly of our detector-cartridge system renders it suitable for quantitative measurements in such teaching activities. Furthermore, in combining hardware, software, physics and biochemistry concepts interdisciplinary teaching activities can be developed for this platform in the future.“

In the results section, I would have liked to read more about the rationale and strategy behind the development of the spectrophotometer. How did you design the device? Did you deconstruct high-tech equipment or model after existing low-cost fluorescence detection equipment? I’d imagine some of the choices made, such as the use of photography lighting filters instead of expensive scientific light filters, might not have been tried before and would be of interest to others seeking to replicate or expand upon this work. Since the design of the fluorescence spectrometer is the crux of this work, I recommend that the authors add discussion of their detector design strategy placing it in the context of previous efforts.

Thank you for pointing this out. We agree that an overall discussion of our design considerations in the context of previous efforts was lacking and added this discussion in the beginning of the results section. We would like to mention, that our final design is quite similar to the design presented by Obahiagbon (2018), we reference accordingly and which was not available when we first presented our concept in 2017 (http://2017.igem.org/Team:Munich/Hardware/Detector).

We now write (line 115):

“The design of our detector is based on the premise that fluorescence produced by biochemical reactions on filter paper should be detected in a most economic (cost as well as power consumption), yet reliable way. The main challenge in building a sensitive detector is maximizing the signal-to-background ratio. To ensure an optimal signal, state of the art laboratory equipment typically uses high-power light sources, focusing optics, and photo multipliers. Background signals are minimized through monochromators, filters, dichroic mirrors and by performing measurements at a 90° or 180° angle relative to the illumination source.

Previous solutions for low-cost detectors avoid some of these requirements, by making use of LEDs for excitation, in combination with photodiodes for detection (Yang et al. 2009, Wu et al. 2012, Novak et al. 2007). However, common 90° or 180° excitation/emission geometries require appropriate optical components, compromising either sensitivity (Yang et al. 2009), or cost (Wu et al. 2012, Novak et al. 2007). They are also difficult to combine with paper-detection formats. 

We hypothesized that measuring at a 0° could allow for a high signal-to-background ratio, while omitting all optical components, except light filters. Since such an arrangement is placing the LED excitation light source and the LDR fluorescence sensor as close as possible (Figure 2a,b). In this setting, the choice of light filters is crucial to not compromise sensitivity, as bleed-through of excitation light to the sensor is the major source of background signal. We thus experimented with cheap photographic light filter foils and found a combination that works sufficiently well in our context. These foils are comparably thin, allowing to reduce the optical path to about 2 mm in total. Conceptually related approaches have been described for using polarizers (Pais et al. 2008) and low-cost bandpass filters (Obahiagbon et al. 2018), respectively.

Our final design features” ...

Minor revisions

A low-cost fluorescence imager designed to enable qualitative or semi-quantitative analysis of cell-free reactions as part of hands-on educational activities was recently described (https://advances.sciencemag.org/content/4/8/eaat5107). Please include this citation in the discussion when educational applications are discussed. 

Thank you very much for this suggestion, we included this very interesting reference and now write:

Conclusions, Line 325: 

“CRISPR/Cas systems and cell-free protein synthesis have become popular tools that are simple to design and operate and have therefore recently been popularized as an educational kit termed BioBits, including a qualitative fluorescence imaging chamber [29, 30].” 

I noticed on the order of 10 typos throughout the manuscript - please address before resubmission. 

We carefully reviewed the text and corrected for several typos.

Reviewer #2: This study (by Katzmeier, Aufinger, Dupen, Quinteiro et al.) develops a low-cost fluorescence reader, and demonstrate applications with Cas13 detection of RNA, and the RNA aptamer Spinach. I think the technology described has promise, but would like to see additional validation to show the sensitivity of Cas13 detection using the device for readout, as well as a few additional controls to validate the performance. With these additions, the paper should be suitable for publication. 

Major comments

 1. Does the cartridge need to be cleaned in between uses? It is unclear if any sample from the filter paper might be carried over between runs. The authors should clarify this point, as it influences the utility of the device. 

To protect the filters from contamination, we covered them with microscopy cover slides. We clean them with ethanol and distilled water between runs. If needed, the filter foils and microscopy cover slides can be replaced within 10 minutes. The schemes in Figure 2 and Figure S1 have been updated to explicitly show the cover slides between the filter foils and the glass fiber paper where the sample is deposited.

We now write (line 169):

“To protect the filter foils from contamination, we covered them with microscopy cover slides for facile cleaning with ethanol and water between measurements.”

Revised Fig 2 detail:

Revised Fig S1:

“Fig S1: a) CAD drawing and b) photograph illustrating the assembly of the assay cartridge. The 3D printed parts are first treated with sand paper. Then the magnets are glued into the corresponding cavities. The filter foils and protective cover slides are cut into the appropriate size, assembled in front of the detection window and fixed with Scotch tape. The filter foils were covered with microscope cover slides to facilitate cleaning with ethanol and water between measurements. To obtain a clean optical pathway, the transmission windows must be free of Scotch tape. A piece of filter paper carrying the sample is placed in front of the transmission window.”

2. The authors should perform a limit of detection assay using Cas13a and their fluorescent detector, if they want to use Cas13 as an application of their device. It is not sufficient to show a single experiment, with 100 nM target, and make claims in the abstract about detection in the "nanomolar range". 

 We answer this comment together with the following comment, as both refer to the sensitivity and the detection limit of the Cas13a assay and of the detector. 

3. Similarly, Table 1 is misleading, as the fluorescein calibration experiments (shown in Fig. 4) appear to be independent of Cas13 - as far as I can tell no sensitivity analysis has been performed with Cas13 (just with fluorescein). 

We agree with the reviewer that our text might have introduced some confusion between the sensitivity of the hardware and the sensitivity of the assay conducted on it. The detection limit stated in the abstract and table 1 exclusively refers to the fluorescence sensitivity of the hardware. We believe that Fluorescein concentration is a reasonably sound and standard metric to judge and compare the performance of the hardware itself. On the other hand, biochemical assays, such as the Cas13a assay, have their own detection limit concerning the minimal amount of active molecule (in our case, target RNA) that induces the assay. The sensitivity of the biochemical assay towards target RNA is different than the sensitivity of the hardware to the read-out molecule (RNase Alert or fluorescein for example). 

The reviewer is of course correct to notice that the sensitivity of the Cas13a assay is important for applications. To see how the sensitivity of the assay compares to the literature when performing the reaction on paper and with our detector, we performed an additional target titration experiment (Figure 6, e and f). 

Fig 6: “e-f) Response of the assay to increasing concentrations of cognate or non-cognate RNA target, in the detector (e) or in bulk (f). The LOD and LOQ as calculated based on the mean and standard deviation of background measurements are shown with dotted lines.”

We note, however, that even a sensitivity of 1 nM RNA is typically not sufficient for any real-world application which usually requires detection limits in the aM range. In the cited references this issue was resolved by pre-amplifying the target RNA with isothermal PCR reactions (for instance RT-RPA-TX), which is performed off-paper and yields RNA in the 100 nM range, as used in our experiments. Such a pre-amplification step is of course compatible with our detector read-out as well.

4. Figure 6b does not have any negative controls, please rerun this experiment with some negative controls (or show them in the plot). 

We thank the reviewer for pointing this out, we have repeated the experiment with a control with 0 nM template: see Figure 5b.

Minor comments

1. Some of the figures could be combined to improve clarity (e.g., 3 and 4; possibly 1 and 2) 

We agree with the reviewer recommendation, and have combined figures 3 and 4:

We preferred to keep figures 1 and 2 separated, as the merged figure contained too much information.

2. What is the final concentration of Cas13a and crRNA in the detection reactions? It is unclear (as written) in the Methods section. 

We updated the corresponding methods section to clearly indicate that the concentrations listed were the final concentrations in the assay. The paragraph "CRISPR/Cas13a-based RNA detection" also specifies the concentrations now.

Line 413:

“For the Cas13a RNA detection assays crRNA was first diluted with processing buffer (1x is 20 mM HEPES pH 6.8, 50mM KCl, 5mM MgCl2 and 5% glycerol) to a final assay concentration of 150 nM crRNA in 1x processing buffer, and incubated for 5min at 65C to ensure complete folding of the crRNA. Then, Cas13a was added to a final assay concentration of 20 nM and the sample was incubated for 10 min at 37C to mediate binding between Cas13a and crRNA. Finally, 200 nM of RNase Alert (Thermo Fischer Scientific), 1.5 µL of RNase Inhibitor, Murine (NEB, No. M0314) and 100 nM of target RNA (all final concentrations in the assay) were added on ice. The solution was mixed by pipetting up and down, and immediately applied to the filter paper and measured.

3. The fluorescence in the presence of noncognate target is increasing over time in panel 7c and 7d, to approximately 1/3 of the maximum signal detected for Cas13 in the presence of cognate target after 1 hour of detection. This is remarkably high. Can the authors comment on why this is the case? Is there some sort of impurity present in the Cas13 or crRNA? This effect does not appear to be specific to the detector system being used, so presumably is coming from one of the reagents.

The reviewer has correctly pointed out the relatively high background activity of the assay in the presence of non-cognate target. We consider this to be a consequence of unspecific Cas13a activity: although Cas13a is activated for RNase activity by target recognition, there is also residual unspecific RNase activity of Cas13a in the absence of target. This is more pronounced in the filter paper assay. Hence, a trade-off must be found between target recognition, where higher concentrations of Cas13a allow for a detection of lower concentrations of target RNA and for faster kinetics, and the unspecific activity of Cas13a, which leads to higher background for higher concentrations of Cas13a. In our optimizations, we identified ~ 20 nM Cas13a as the optimal concentration for a high signal-to-noise ratio, consistent with previous literature reports.

Furthermore, target RNA induced signal saturated already after 40 min, at which point the assay would be stopped while non-cognate target still reports low fluorescence values. We choose to plot the full hour recording however to point the reader towards this common assay artefact. For further improving clarity, we now mentioned the unspecific activity of Cas13a in the legend of the revised new Fig 6c-f:

“ ... c-d) Measurement of Cas13a activity with 100 nM target RNA on paper in the detector (c) or in bulk using a plate reader (d). Activity in presence of a cognate (i.e. complementary to the crRNA) RNA target is compared to a non-cognate target. Residual activity in the presence of non-cognate target is likely due to an unspecific activity of Cas13a. In d), positive control in bulk contains RNase A.“...

4. There appears to be a broken reference (a "?" in line 72). Please fix, and proofread the manuscript to ensure that no other such typos are present. 

Thank you for noticing this typo, we corrected the mistake and carefully proofread the manuscript.

5. The authors reference several point of care technologies, but have omitted a recent demonstration of Cas13 detection of viruses at the point of care, which was published alongside the SHERLOCK 2.0 paper: doi: 10.1126/science.aas8836 

Thank you for directing us to this very interesting reference, which we now include (Myhrvold 2018).

Reviewer #3: In this manuscript, the authors developed a low-cost fluorescence reader for POCT application. With the employment of the CRIPSR-Cas13 collateral cleavage activity against RNA, target RNA can be conveniently detected by the developed reader with high sensitivity. And this invent surely has the potential to be widely used in both clinical and household application scenarios. I have only several minor concerns for the authors to address before the work can be accepted for publication. 

1) The filters can be polluted by the paper strip. How to avoid the pollution?

Reviewer 2 raised a similar concern. Please find our detailed response to his/her comment 1.

2) In figure 7, longer reaction time is required for the detector, and the authors proposed several possibilities. However, it is still highly recommended that the authors may test with increased amount of enzyme with the detector. 

The concentration of Cas13a has been optimized at 20 nM to mitigate two opposite effects: on the one hand, a higher Cas13a concentration allows, as the reviewer points out, faster kinetics and for a detection of lower concentrations of target RNA. On the other hand, Cas13a has unspecific activity, meaning some low RNAse activity can be detected in the absence of crRNA and target RNA. This effect is more significant with increasing concentrations of Cas13a. For this reason, we abstained from using higher concentrations of the enzyme in the detector experiment. We also specified in the caption of Figure 6 that the measured increase in fluorescence with non-cognate target is likely due to the unspecific RNAse activity of Cas13a.

3) To demonstrate the practicability of the device and the system, the authors may need to detect a real clinical or nonclinical sample with the system. 

We concur, that such data would be highly desirable. Sadly, we do not have permission or access to clinical samples and national regulations require stringent vetting and patient consent prior to any such experiments, which is beyond the capacity of our undergraduate student team to complete. We therefore hope that this publication with a focus on the fluorescence reader will nevertheless be of interest to the community.

4) Besides of Cas13, Cas12 has also been used for CRISPR-Dx (i.e. HOLMES and DETECTR), and the authors may need to discuss them or at least describe in the introduction part. If the reader is also compatible with the Cas12 system, more readers will be interested. 

We thank the reviewer for suggesting these interesting references and have included them in the introduction.

---

## [Decision Letter · Decision Letter 1]

23 Oct 2019

A low-cost fluorescence reader for in vitro transcription and nucleic acid detection with Cas13a

PONE-D-19-18880R1

Dear Dr. Heymann,

We are pleased to inform you that your manuscript has been judged scientifically suitable for publication and will be formally accepted for publication once it complies with all outstanding technical requirements.

**Please note one outstanding point from Reviewer #2****, which should be amended at this final submission stage:**

"One key component is still missing: the make, model, and settings used to acquire fluorescence data with the plate reader. This will allow the reader to more easily interpret and compare results between the author's device and a more standard setup, and is important for reproducibility."

With kind regards,

Mark Isalan

Section Editor

PLOS ONE

Additional Editor Comments (optional):

Reviewers' comments:

Reviewer's Responses to Questions

**Comments to the Author**

1. If the authors have adequately addressed your comments raised in a previous round of review and you feel that this manuscript is now acceptable for publication, you may indicate that here to bypass the “Comments to the Author” section, enter your conflict of interest statement in the “Confidential to Editor” section, and submit your "Accept" recommendation.

Reviewer #1: All comments have been addressed

Reviewer #2: (No Response)

Reviewer #3: All comments have been addressed

2. Is the manuscript technically sound, and do the data support the conclusions?

Reviewer #1: Yes

Reviewer #2: Yes

Reviewer #3: Yes

3. Has the statistical analysis been performed appropriately and rigorously? 

Reviewer #1: N/A

Reviewer #2: Yes

Reviewer #3: N/A

4. Have the authors made all data underlying the findings in their manuscript fully available?

Reviewer #1: Yes

Reviewer #2: Yes

Reviewer #3: Yes

5. Is the manuscript presented in an intelligible fashion and written in standard English?

Reviewer #1: Yes

Reviewer #2: Yes

Reviewer #3: Yes

6. Review Comments to the Author

Reviewer #1: The authors have very satisfactorily addressed all of my concerns from the initial round of review. I recommend to accept the revised manuscript.

Reviewer #2: The authors have addressed nearly all of my comments. One key component is still missing: the make, model, and settings used to acquire fluorescence data with the plate reader. This will allow the reader to more easily interpret and compare results between the author's device and a more standard setup, and is important for reproducibility.

Reviewer #3: The revised manuscript has been much improved and all of my previous concerns have been fully addressed. Considering the fierce competition in the field of CRISPR diagnostics, this referee suggests that this work should be accepted for publishing on PLOS ONE without much delay.

7. PLOS authors have the option to publish the peer review history of their article (what does this mean?). If published, this will include your full peer review and any attached files.

Reviewer #1: No

Reviewer #2: No

Reviewer #3: No

---

## [Editor Report · Acceptance letter]

14 Nov 2019

PONE-D-19-18880R1 

A low-cost fluorescence reader for in vitro transcription and nucleic acid detection with Cas13a 

Dear Dr. Heymann:

I am pleased to inform you that your manuscript has been deemed suitable for publication in PLOS ONE. Congratulations! Your manuscript is now with our production department. 

With kind regards,

on behalf of

Dr. Mark Isalan 

Section Editor

PLOS ONE